# Factors Related to Caregivers’ Intention to Vaccinate Their Elderly Family Members with Major Neurocognitive Disorders against COVID-19

**DOI:** 10.3390/vaccines12060668

**Published:** 2024-06-17

**Authors:** Mei-Feng Huang, Yi-Chun Yeh, Tai-Ling Liu, Ray C. Hsiao, Cheng-Sheng Chen, Cheng-Fang Yen

**Affiliations:** 1Department of Psychiatry, Kaohsiung Medical University Hospital, Kaohsiung Medical University, Kaohsiung 80754, Taiwan; lalalabon@gmail.com (M.-F.H.);; 2Department of Psychiatry, School of Medicine, College of Medicine, Kaohsiung Medical University, Kaohsiung 80708, Taiwan; 3Department of Psychiatry, Seattle Children’s, Seattle, WA 98195, USA; 4Department of Psychiatry and Behavioral Sciences, School of Medicine, University of Washington, Seattle, WA 98105, USA; 5College of Professional Studies, National Pingtung University of Science and Technology, Pingtung 91201, Taiwan

**Keywords:** caregiver, major neurocognitive disorder, vaccine, COVID-19

## Abstract

Vaccination helps reduce the risk of coronavirus disease 2019 (COVID-19) infection in elderly individuals with major neurocognitive disorders (MNDs). However, some caregivers are hesitant to have their elderly family members with MNDs vaccinated against COVID-19. This study explored the factors influencing caregivers’ intentions to vaccinate elderly family members with MNDs against COVID-19. A total of 232 caregivers of elderly family members with MNDs participated in this study. In this survey, data regarding COVID-19 vaccination acceptance, fear, side effects, family members’ attitudes toward vaccination, mental health status, neuropsychiatric symptoms, and cognitive impairments were collected from the elderly participants with MNDs. The associations between these variables and the caregivers’ intention to vaccinate their elderly family members with MNDs against COVID-19 were examined using a multivariable linear regression analysis model. The results revealed that caregivers’ perceived familial support for vaccination, the perceived value of vaccination, and autonomy to vaccinate elder family members were positively correlated with caregivers’ intention to vaccinate elderly family members with MNDs, whereas elderly family members’ age was negatively correlated with caregiver intentions. This study demonstrated that caregiver factors (perceived familial support, value of vaccination, and autonomy) and elderly family members’ age were correlated with caregiver intention. These factors should be considered in developing interventions to enhance caregivers’ intentions to vaccinate their elderly family members with MNDs against COVID-19.

## 1. Introduction

The coronavirus disease 2019 (COVID-19) emerged in late 2019 and brought great harm to the world. Although the World Health Organization declared on 3 May 2023 that COVID-19 was no longer a public health emergency of international concern, the numbers of morbidities and mortalities due to the contraction of COVID-19 continue to increase. A total of 775,293,630 confirmed cases of COVID-19 worldwide had been reported as of 26 April 2024 [1]. Excess mortality rates exceeded 150 deaths per 100,000 people during at least one year of the COVID-19 pandemic in 80 countries and territories [2]. Elderly individuals have been identified as a population with a high mortality rate of COVID-19, severe COVID-19 symptoms and complications, and the need for invasive mechanical ventilation [3]. In particular, elderly individuals with the diagnosis of a major neurocognitive disorder (MND) have increased COVID-19 severity and mortality rates [4,5,6,7,8,9,10,11]. A study found that large increases in mortality in individuals with MNDs such as Alzheimer’s disease as an underlying or contributing cause of death occurred in the first year of the COVID-19 pandemic in the United States [12]. In addition to old age and comorbidities [8], ApoE4, a strong genetic risk factor for Alzheimer’s disease, has been associated with an increased risk of severe COVID-19 [6]. These vulnerabilities make it especially important to minimize the exposure of people with MNDs to COVID-19 [5].

Vaccination is a crucial strategy to mitigate the risk of contracting COVID-19 and related hospitalizations [13]. Studies have confirmed the effectiveness of vaccines against COVID-19 among elderly individuals with MNDs [14,15]. For example, a prospective cohort study analyzed the medical records of 25,733 elderly individuals with MNDs in Israel and found that the mortality rate of COVID-19 was 52% for unvaccinated individuals and 7% for vaccinated individuals [15]. Studies have also supported the use of COVID-19 vaccines as safe for elderly individuals with MNDs, who should be prioritized in the vaccination campaign [16,17]. A study on elderly individuals with MNDs did not detect an increased risk of delirium or other adverse events following vaccination compared to the prepandemic period and those who tested positive for COVID-19 [16]. Another study also found that cognitive and behavioral changes following vaccination were rarely reported in elderly individuals with MNDs [16]. However, studies have found that some caregivers are hesitant to have their elderly family members with MNDs vaccinated against COVID-19 [18,19]. A study in Shanghai, China, from April to May 2022 found that only 27.1% of elderly individuals with MNDs living in community-dwelling settings were vaccinated against COVID-19 [20]. Caregivers played a crucial role in taking care of elderly individuals with MNDs during the COVID-19 pandemic. Although the World Health Organization declared on 3 May 2023 that COVID-19 was no longer a public health emergency of international concern, the numbers of confirmed COVID-19 cases and COVID-19-associated deaths worldwide have continued to increase. Therefore, examining the factors related to caregivers’ intention to vaccinate their elderly family members with MNDs is essential to the development of intervention programs for enhancing the rate of vaccination in elderly individuals with MNDs.

Several factors have been found to be associated with caregivers’ intention to vaccinate their elderly family members with MNDs. A qualitative study found that perceived direct and indirect health risks of contracting COVID-19, information regarding vaccination from trusted people, and the ability to overcome vaccination barriers affected caregiver COVID-19 vaccination acceptance, whereas caregiver vaccine refusals were motivated by a low perceived risk of COVID-19, vaccine fear, and personal beliefs [18]. A questionnaire survey study found that caregivers’ concern regarding the aggravation of elderly family members’ health conditions by vaccination, negative reports about the vaccination, and adverse reactions to vaccination were related to caregiver vaccine hesitancy [19]. Pain at the injection site, fever, and fatigue were the most common side effects in people receiving COVID-19 vaccines [21]. Major adverse effects of COVID-19 vaccines, such as cardiovascular, neurologic, hematologic, and immune-allergic side effects, have also been reported [22]. For example, several vaccine-induced thrombotic thrombocytopenia syndrome cases have been reported after the ChAdOx1 nCov-19 vaccination [23]. Research also found that elderly individuals with MNDs who were younger, had higher cognitive and daily functions, and had no neuropsychiatric symptoms of agitation and aggression were more likely to receive the COVID-19 vaccination, whereas caregivers who had a lower overall caregiver burden and burden of frustration were more likely to vaccinate their elderly family members with MNDs [20].

Researchers have employed both the theory of planned behavior (TPB) [24] and protection motivation theory (PMT) [25] to investigate the factors influencing individuals’ intention to receive vaccination against COVID-19. According to the TPB, three main factors influence individuals’ intention to receive vaccination against COVID-19: attitude (an individual’s overall evaluation of whether vaccination is favorable or unfavorable), subjective norm (the perceived social pressure to receive the vaccination), and perceived behavioral control (the individual’s belief in their ability to receive vaccination) [24,26]. The PMT posits that the perceived severity of the health threat caused by COVID-19 and vulnerability to COVID-19 (threat appraisals) and the perceived efficacy of vaccination in alleviating the threat of COVID-19 (coping appraisals) influence individuals’ intention to receive vaccination against COVID-19 [27,28,29,30,31]. Moreover, the emergence of social media as a vital platform for disseminating COVID-19 vaccine information has altered information processing patterns [32], despite the prevalence of health misinformation across the majority of social media platforms [33]. Concerns regarding vaccine side effects also contribute to the hesitancy toward COVID-19 vaccines [34]. Moreover, caregiver mental health may influence their decisions regarding vaccination for their families [35]. However, a comprehensive examination of the factors influencing caregivers’ intention to vaccinate their elderly family members with MNDs against COVID-19 has yet to be conducted.

The objective of the present study was to investigate the associations of various factors with caregivers’ intention to vaccinate their elderly family members with MNDs against COVID-19. Specifically, we hypothesized that caregivers have greater intention to vaccinate their elderly family members with MNDs against COVID-19 when they have a greater fear of COVID-19, have a stronger perception of the value of the COVID-19 vaccination for their elderly family members’ health, have a greater awareness of the positive impacts of vaccination on the health of their elderly family members, have greater knowledge regarding vaccination, have greater autonomy in decision-making regarding vaccination, have fewer concerns regarding the side effects of vaccination for their elderly family members, perceive more favorable attitudes from other family members toward vaccinating their elderly family members, have greater exposure to information on COVID-19 vaccines from social media, and have better mental health. We also hypothesized that higher cognitive and daily functions in elderly family members with MNDs are positively correlated with caregivers’ intentions.

## 2. Methods

### 2.1. Participants

The study participants comprised the primary caregivers of elderly individuals with a diagnosis of an MND, enrolled from six psychogeriatric outpatient clinics in two hospitals in Taiwan between February and April 2024. The inclusion criteria for primary caregivers were as follows: (1) aged 20 years or older, and (2) primary caregivers of elderly individuals aged 60 years or older and who had received a diagnosis of an MND by a psychiatrist in accordance with the fifth edition of the *Diagnostic and Statistical Manual of Mental Disorders (DSM-5)* [36]. Caregivers with impaired intellect or cognitive problems due to a head injury, major physical or psychiatric problems, or substance use problems were excluded from the present study. Psychiatrists told caregivers, “This is a questionnaire study to understand caregivers’ intention and correlates to have their elderly family members with MND vaccinated against COVID-19. The results of the study will serve as a reference for clinicians to develop strategies to enhance caregivers’ efforts to have their elderly family members with MND vaccinated. The results of the questionnaire are confidential. You can be assured that if you do not want to participate in the study, it will not affect the medical rights of your family members.” Participants were assured that their responses were confidential and that their participation or nonparticipation would not influence the right of their family members with MNDs to receive medical services. Subsequently, we consecutively invited 232 caregivers of elderly individuals with MNDs into this study. According to Green [37], the total number of participants in studies using a regression analysis model needs to be at least 50 + 8 × (the number of independent variables). There were 16 independent variables in this study; this study needed at least 178 participants. Therefore, 232 participants were enough for the regression analysis used in this study.

### 2.2. Ethics Statement

The present study was approved by the institutional review board of Kaohsiung Medical University Hospital (protocol number, KMUH-E(I)-20240044; date of approval, 26 January 2024). Participants provided written informed consent for their involvement. The study adhered to the principles of the Declaration of Helsinki and the guidelines for the Conduct, Reporting, Editing, and Publication of Scholarly Work in Medical Journals.

### 2.3. Measures

#### 2.3.1. Caregiver Intention to Vaccinate Elderly Family Members with MNDs against COVID-19

In this study, caregivers were asked about their intention to vaccinate their elderly family members with an MND against COVID-19 by using the following question: “Please rate your current willingness to let your elderly families with MND receive COVID-19 vaccination.” Responses were rated on a Likert scale ranging from 1 (very low) to 10 (very high) [38].

#### 2.3.2. Caregiver Motors to Vaccinate Their Elderly Family Members with MNDs against COVID-19

The present study employed the 12-item caregiver version of the Motors of COVID-19 Vaccination Acceptance Scale (P-MoVac-COVID19S-12) [39] to assess caregiver motivation to vaccinate their elderly family members with MNDs against COVID-19. The P-MoVac-COVID19S-12 comprises four domains, including values of vaccination, impacts caused by vaccination, knowledge of vaccines, and autonomy to vaccinate families. Participants responded to all items using a 7-point scale, with endpoints ranging from 1 (strongly disagree) to 7 (strongly agree). A cumulative higher score in each domain suggests increased caregiver motivation to vaccinate their elderly family members with MNDs against COVID-19. Cronbach’s α values of the four domains of P-MoVac-COVID19S-12 in this study ranged between 0.74 and 0.82.

#### 2.3.3. Caregiver Concerns Regarding the Side Effects of COVID-19 Vaccines

The caregiver’s level of concern regarding the potential side effects of COVID-19 vaccines for their elderly family members with MNDs was assessed using the following question: “How concerned are you about the possible side effects of COVID-19 vaccine for your elderly families with MND?” The caregivers rated their level of concern on a 4-point scale ranging from 0 (not at all) to 3 (extremely).

#### 2.3.4. Other Family Members’ Attitudes toward Vaccination of Elderly Family Members with MNDs against COVID-19

The level of other family members’ attitudes toward vaccination of elderly family members with MNDs against COVID-19 perceived by caregivers was assessed using the following question: “What is the level of support from other family members regarding your views on having this elderly family with MND vaccinated against COVID-19?” Caregivers rated their responses on a 4-point scale from 1 (very low) to 4 (very high).

#### 2.3.5. Obtaining Information about COVID-19 Vaccines from Social Media

We asked the caregivers how frequently they obtained information about COVID-19 vaccines from social media, including Facebook, LINE, Instagram, and Twitter. Potential responses were never, seldom, sometimes, and frequently. Caregivers who responded sometimes or frequently were classified as having obtained information on COVID-19 vaccines from social media.

#### 2.3.6. Caregiver Mental Health Status

Caregivers’ mental health status in the last month was assessed by using the 5-item Brief Symptom Rating Scale (BSRS-5) [40]. The BSRS-5 assesses caregivers’ self-reported mental health indicators, including feeling tense or keyed up (anxiety), feeling low in mood (depression), feeling easily annoyed or irritated (hostility), feeling inferior to others (interpersonal hypersensitivity), and having trouble falling asleep (insomnia). Caregivers rated each item on a 5-point scale ranging from 0 (*not at all*) to 4 (*extremely*). A higher total BSRS-5 score indicates poorer mental health. A total BSRS-5 score of 6 or higher indicates a poor mental health status [40]. The BSRS-5 has been demonstrated to possess satisfactory psychometric properties, making it a reliable measure for detecting psychiatric morbidity both in medical settings and community contexts [40]. In this study, Cronbach’s α coefficient for the BSRS-5 was 0.88.

#### 2.3.7. Caregiver Fear of COVID-19

Caregivers rated their level of fear of COVID-19 on the Fear of COVID-19 Scale (FCV-19S) [41]. Ratings were given on a 5-point Likert scale ranging from 1 (*strongly disagree*) to 5 (*strongly agree*). An analysis of several studies has independently validated the psychometric properties of FCV-19S obtained in samples of a generally reasonable size in diverse populations [42]. The Taiwanese version of the FCV-19S has satisfactory validity and reliability [43]. Cronbach’s α of the FCV-19S in this study was 0.91.

#### 2.3.8. Neuropsychiatric Symptoms of Elderly Family Members with MNDs

The presence and severity of neuropsychiatric symptoms were assessed using the Neuropsychiatric Inventory Questionnaire (NPI-Q) [44]. The NPI-Q contains 12 items assessing individuals’ delusions, hallucinations, agitation/aggression, depression/dysphoria, anxiety, apathy/indifference, euphoria/elation, irritability/lability, disinhibition and aberrant motor activity, eating habit changes, and sleep problems. Each item was rated on a 4-point scale ranging from 0 (*no symptoms*) to 3 (*severe*). A higher total NPI-Q score indicates more severe neuropsychiatric symptoms. Cronbach’s α value of the NPI-Q in this study was 0.88.

#### 2.3.9. Cognitive Impairments of Elderly Family Members with MNDs

The cognitive impairments of elderly family members with MNDs were assessed using the Clinical Dementia Rating Scale (CDR) [45]. Psychiatrists collected information from both the elderly individuals with MNDs and their caregivers and evaluated the elderly individuals’ six domains of cognitive impartments due to their MND, including memory, orientation, judgment and problem-solving, community affairs, home and hobbies, and personal care. Psychiatrists rated the severity of cognitive impairments for each CDR domain on a 5-point scale (except for the personal care domain) and then synthesized them to assign a global CDR score.

#### 2.3.10. Demographics of Caregiver and Elderly Individuals with MNDs

Information regarding the gender, age, and educational level (high school or below vs. college or above) of the caregivers and the gender and age of the elderly individuals with MNDs in their care was collected.

### 2.4. Statistical Analysis

Statistical analyses were performed using IBM SPSS Statistics version 24.0 (IBM Corporation, Armonk, NY, USA). The demographics, caregivers’ intention and motors to vaccinate elderly family members against COVID-19, concerns regarding the side effects, other family members’ perceived attitudes, obtaining information from social media, mental health status and fear of COVID-19, and elderly family members with MNDs’ neuropsychiatric symptoms and cognitive impairments are presented as means (standard deviations) and frequencies (percentages). To determine the normal distribution of continuous variables, criteria of absolute values of <7 and <3 for kurtosis and skewness, respectively, were applied [46]. These tests did not reveal significant deviations. The present study used bivariable linear regression analysis to examine the associations between independent factors and caregivers’ intention to vaccinate elderly family members with MNDs against COVID-19. Factors significantly associated with caregiver intention were further analyzed in a multivariable linear regression model. A two-tailed *p* value of <0.05 was considered statistically significant.

## 3. Results

A total of 232 caregivers (143 women and 89 men) participated in this study. Table 1 presents the characteristics of the participants and the characteristics and scores of CDR and NPI-Q of their elderly family members with MNDs. The mean age of the participants was 58.1 ± 11.3 years. Nearly three-fifths (59.9%) of the caregivers had attained an education degree from college or above, with 25.4% reporting poor mental health status. Regarding the elderly family members with MNDs (159 were women and 73 were men), the mean age of their elderly family members with MNDs was 81.5 ± 8.7 years. The most common diagnosis was Alzheimer’s disease (73.7%), followed by vascular dementia (11.6%). Their mean severity of neuropsychiatric symptoms on the NPI-Q was 11.2 ± 7.6, and the mean severity of cognitive impairments on the CDR was 1.5 ± 0.5. Table 2 presents the caregivers’ intention to vaccinate their elderly family members with MNDs, concerns regarding the side effects of COVID-19 vaccines, perceived familial support for vaccinating elderly family members with MNDs, received information about COVID-19 vaccines from social media, and the scores of the P-MoVac-COVID19S-12 and FCV-19S.

Table 3 presents the results of the bivariable and multivariable linear regression analyses examining the factors related to caregivers’ intention to vaccinate their elderly family members with MNDs against COVID-19. Factors that were significantly correlated with caregivers’ intention to vaccinate their elderly family members with MNDs against COVID-19 in bivariable linear regression analysis models (caregivers’ perceived familial support for vaccinating, four domains of motors to vaccinate elderly family members with MNDs, concern regarding the side effects of COVID-19 vaccines, and elderly family members’ age) were further entered into a multivariable linear regression analysis model (Table 3). The analysis revealed that caregiver-perceived familial support for vaccination (*p* < 0.001), the perceived value of vaccination (*p* = 0.002), and autonomy to vaccinate elderly family members with MNDs (*p* = 0.001) were positively correlated with caregivers’ intention to vaccinate elderly family members with MNDs, whereas elderly family members’ age was negatively correlated with caregivers’ intention (*p* = 0.036). The variance inflation factor of these seven variables ranged between 1.021 and 4.171, and the condition index was 29.623. The results indicated no collinearity, as determined on the basis of the suggestions of Senaviratna and Cooray [47].

## 4. Discussion

The present study demonstrated that caregiver-perceived familial support for vaccination, the perceived value of vaccination, and autonomy to vaccinate elderly family members were positively correlated with caregivers’ intention to vaccinate elderly family members with MNDs, whereas elderly family members’ age was negatively correlated with caregiver intention.

In this study, caregivers expressed a moderate to high level of intention to vaccinate their elderly family members with MNDs against COVID-19, with a mean score of 7.1 on a 10-point Likert scale. However, some caregivers remained hesitant to have their elderly family members with MNDs vaccinated against COVID-19, as evidenced by 24 (10.3%) participants, who rated their intention to vaccinate as very low (score of one). Considering the benefits of the COVID-19 vaccination, educating caregivers about its importance and strengthening their intention to have their elderly family members with MNDs vaccinated is essential.

The present study found that caregiver-perceived familial support for the vaccination of elderly family members with MNDs against COVID-19 was positively correlated with caregivers’ intentions. A previous study also found that perceived unfavorable family attitudes toward vaccinating their children were correlated with caregiver hesitancy to vaccinate their children [38]. According to the TPB [24,26], the favorable opinions of other family members can give caregivers a sense of the norm of the attitudes of those around them toward vaccination and reduce caregivers’ hesitation to have elderly family members with MNDs vaccinated against COVID-19. According to ecological systems theory [48], caring for elderly family members with MNDs causes issues in interaction among caregivers, elderly family members with MNDs, and other family members. Family members who support vaccination can also provide vaccination information to caregivers and get elderly family members with MNDs vaccinated as soon as possible. Information regarding vaccination from trusted people also affects caregiver COVID-19 vaccination acceptance [18]. This suggests that the attitudes of both the primary caregiver and other family members may influence whether or not elderly individuals with MNDs receive vaccination in a society with strong family ties. Healthcare professionals should introduce the value of vaccination against COVID-19 to as many family members as possible, not just primary caregivers.

The present study found that two caregiver motors of vaccination (including the perceived value of vaccination and autonomy to vaccinate elderly family members) were positively correlated with caregivers’ intention to vaccinate elderly family members with MNDs. According to the cognitive model of empowerment [49,50], the value motor of vaccination indicates how much the caregivers care about the purpose of vaccination uptake, and the autonomy motor indicates how much confidence and control the caregivers have in getting vaccinated if they are willing to. Caregivers who have greater motors of value and autonomy are able to proactively search for resources to obtain vaccinations and develop strategies to remove barriers to vaccination for their elderly family members with MNDs. The present study also found that the positive impact motor of vaccination tended to be positively correlated with caregivers’ intention to vaccinate elderly family members with MNDs. The items on the impact domain of the P-MoVac-COVID19S-12 asked caregivers for their agreement with the effects of vaccination on protecting elderly family members with MNDs against COVID-19 and reducing their risk of catching COVID-19 [39]. According to the PMT [27,28,29,30,31], the perceived efficacy of vaccination in alleviating the threat of COVID-19 influences individuals’ intention to receive vaccination against COVID-19. Caregivers who perceive greater impacts have stronger beliefs in the differences made by vaccination uptake to prevent COVID-19 transmission; thus, their intention to vaccinate their elderly family members with MNDs increases. Based on the results of this study, healthcare professionals can pave the way to target caregivers’ vaccine hesitancy by designing bespoke and potentially effective interventions to enhance their intention to vaccinate elderly family members with MNDs.

The present study found that caregivers had lower intentions to vaccinate older family members with MNDs against COVID-19. The result was congruent with that of a previous study [20]. Caregivers may be concerned that older family members with MNDs may not be able to tolerate the side effects of vaccines and thus have a lower intention to have them vaccinated against COVID-19. However, it is well established that the use of COVID-19 vaccines is safe for elderly individuals with MNDs [16,17]. Because of the increased COVID-19 severity and mortality in this group, elderly individuals with MNDs should be prioritized in the vaccination campaign.

The present study did not find significant associations between caregiver concerns regarding the side effects of COVID-19 vaccines, fear of COVID-19, and individuals with MNDs’ cognitive dysfunctions and neuropsychiatric symptoms and caregiver intention. The results of the present study were not congruent with those of the previous studies [18,19,20]. Studies have found that high caregiver concerns regarding the side effects of COVID-19 vaccines and low caregiver fear of COVID-19 have been found to be correlated with low caregiver intention to vaccinate their elderly family members with MNDs [18,19]. High cognitive and daily dysfunctions and low neuropsychiatric symptoms of agitation and aggression have also been found to be correlated with vaccination against COVID-19 in elderly individuals with MNDs [20]. The present study invited caregivers from outpatient clinical units whose elderly family members with MNDs were currently undergoing treatment, and the caregivers could receive accurate information regarding COVID-19 and vaccines from physicians. Moreover, elderly individuals with MNDs might receive medication to slow cognitive deterioration and reduce neuropsychiatric symptoms. Thus, the associations of these caregiver and patient factors with caregiver intention in this clinical sample may differ from those in the community. Further studies are needed to replicate the results of this study.

Social media was one of the main information sources for COVID-19 vaccines during the COVID-19 pandemic [32], though misinformation is prevalent on social media platforms [33]. In this study, the association between receiving the COVID-19 vaccine, information from social media, and caregiver intention to vaccinate their elderly family members with MNDs was nonsignificant. It is possible that caregivers could obtain vaccine information from physicians in medical units and reduce their reliance on messages from social media. Studies have found that caregivers with high caregiver burden and frustration had a low intention to vaccinate their elderly family members with MNDs [20]. Caregivers’ burden and frustration may compromise caregivers’ mental health and reduce their ability to plan to vaccinate their families [35]. However, the present study did not find a significant association between caregiver mental health status and caregiver intention to vaccinate their elderly family members with MNDs. Further studies are needed to replicate the results of this study.

### Strengths and Limitations

The present study is the first to comprehensively examine the roles of caregiver vaccination behaviors, attitudes, practices, sources of information, mental health status, and elderly individuals’ neuropsychiatric symptoms and cognitive impairments on caregivers’ intention to vaccinate elderly family members with MNDs against COVID-19. The results of this study provide evidence for developing intervention programs to enhance caregivers’ intention to vaccinate their elderly family members with MNDs against COVID-19. This study has several limitations. First, the data collection relied solely on self-reports from caregivers of elderly individuals with MNDs, potentially introducing single-informant bias and social desirability bias. Second, this study examined the cross-sectional associations between the factors and caregivers’ intentions and thus did not establish causality. Third, whether this study’s results can be generalized to caregivers of elderly individuals with MNDs who have not sought medical assistance in outpatient settings remains uncertain. Further studies on caregivers of elderly individuals with MNDs in the community are needed to replicate the results of this study. Fourth, caregivers of elderly individuals without MNDs were not recruited in this study for comparison. Further studies are warranted to examine whether the factors related to caregivers’ intentions identified in this study also relate to the intentions of caregivers of elderly individuals without MNDs. Fifth, the participants may also have provided socially desirable responses instead of choosing responses reflective of their true feelings. Potential information bias should be examined in further studies. Sixth, this study was conducted in 2024, which was well past the peak of COVID-19. However, as large-scale respiratory viral infectious disease outbreaks are still possible in the future, continued research is still necessary.

## 5. Conclusions

This study discovered that caregiver-perceived familial support for vaccination, the perceived value of vaccination, autonomy to vaccinate elderly family members, and elderly family members’ age were correlated with caregivers’ intention to vaccinate elderly family members with MNDs. Based on the results of this study, healthcare professionals should formulate strategies to enhance the intention among caregivers to vaccinate their elderly family members with MNDs against COVID-19. Priority should be given to intervention programs targeting caregivers who have older family members with MNDs. Healthcare professionals should understand caregivers’ perceived attitudes of other family members toward vaccination and assist them in shaping the ability to communicate the necessity of vaccination to other family members. Moreover, these findings underscore the requirement for programs designed to enhance caregivers’ motors of vaccination, such as value, positive impacts, and autonomy, to increase their intention to vaccinate their elderly family members with MNDs against COVID-19. Further studies are needed to examine caregivers’ intentions and related factors to vaccinate elderly family members with MNDs in the community population. The sources of information regarding COVID-19 and vaccination also warrant examination.

## Figures and Tables

**Table 1 vaccines-12-00668-t001:** Characteristics of caregivers and elderly family members with major neurocognitive disorders.

	*n* (%)	Mean (SD)	Range
Caregiver (*n* = 232)			
Gender			
Women	143 (61.6)		
Men	89 (38.4)		
Age (year)		58.1 (11.3)	24–94
Education level			
High school or below	93 (40.1)		
College or above	139 (59.9)		
Mental health status on the BSRS-5	173 (74.6)		
Good to fair	59 (25.4)		
Poor			
Elderly family members with MNDs (*n* = 232)			
Gender			
Women	159 (68.5)		
Men	73 (31.5)		
Age (year)		81.5 (8.7)	60–102
Diagnosis of MND			
Alzheimer’s disease	171 (73.7)		
Vascular dementia	27 (11.6)		
Dementia with Lewy bodies	16 (6.9)		
Frontotemporal dementia	7 (3.0)		
Others	11 (4.7)		
Neuropsychiatric symptoms on the NPI-Q		11.2 (7.6)	0–34
Severity of cognitive impairments on the CDR		1.5 (0.5)	1–3

BSRS-5: 5-item Brief Symptom Rating Scale; CDR: Clinical Dementia Rating Scale; MNDs: major neurocognitive disorders; NPI-Q: Neuropsychiatric Inventory Questionnaire.

**Table 2 vaccines-12-00668-t002:** Caregiver vaccination behaviors, attitudes, and practices (*N* = 232).

	*n* (%)	Mean (SD)	Range
Intention to vaccinate elderly family members with MNDs		7.1 (3.1)	1–10
Motors of vaccinating elderly family members with MNDs			
Impact		16.9 (3.6)	5–21
Knowledge		15.0 (3.7)	5–21
Value		17.0 (3.6)	5–21
Autonomy		14.0 (3.8)	3–21
Fear of COVID-19 on the FCV-19S		17.3 (6.3)	7–34
Concern regarding side effects of COVID-19 vaccines		1.4 (0.9)	0–3
Perceived familial support for vaccinating elderly family members with MNDs		3.0 (1.0)	1–4
Receiving COVID-19 information from social media	109 (47.0)		

FCV-19S: Fear of COVID-19 Scale; MNDs: major neurocognitive disorders.

**Table 3 vaccines-12-00668-t003:** Factors related to caregiver intention to vaccinate elderly family members with major neurocognitive disorders against COVID-19: bivariable and multivariable linear regression analysis.

	Bivariable Linear Regression		Multivariable Linear Regression	
	B (se)	*p*	B (se)	*p*
Caregivers’ gender ^a^	0.717 (0.422)	0.091		
Caregivers’ age	−0.020 (0.018)	0.282		
Caregivers’ education level ^b^	−0.427 (0.420)	0.311		
Caregivers’ fear of COVID-19	−0.043 (0.033)	0.195		
Caregivers’ concern regarding side effects of COVID-19 vaccines	−0.942 (0.215)	<0.001	−0.028 (0.157)	0.857
Caregivers’ perceived family support for vaccinating elderly family members with MNDs	2.006 (0.148)	<0.001	1.260 (0.152)	<0.001
Caregivers’ mental health status ^c^	−0.395 (0.473)	0.405		
Motors of vaccinating elderly family members with MNDs: impact	0.522 (0.046)	<0.001	0.141 (0.073)	0.054
Motors of vaccinating elderly family members with MNDs: knowledge	0.418 (0.049)	<0.001	−0.062 (0.052)	0.238
Motors of vaccinating elderly family members with MNDs: value	0.531 (0.044)	<0.001	0.226 (0.073)	0.002
Motors of vaccinating elderly family members with MNDs: autonomy	0.385 (0.049)	<0.001	0.146 (0.042)	0.001
Caregivers receiving COVID-19 information from social media	−0.372 (0.413)	0.368		
Elderly family members’ gender ^a^	0.052 (0.445)	0.906		
Elderly family members’ age	−0.062 (0.023)	0.009	−0.032 (0.015)	0.036
Neuropsychiatric symptoms	−0.031 (0.027)	0.254		
Cognitive impairments	−0.341 (0.328)	0.299		

^a^: women as the reference. ^b^: high school or below as the reference. ^c^: good or fair status as the reference. MNDs: major neurocognitive disorders.

## Data Availability

The data are available upon reasonable request to the corresponding authors.

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
