# Peer review of "Factors Related to Caregivers’ Intention to Vaccinate Their Elderly Family Members with Major Neurocognitive Disorders against COVID-19"

_vaccines, 2024, doi:10.3390/vaccines12060668_

Round 1

Reviewer 1 Report

Comments and Suggestions for Authors

The abstract is nicely written. I feel that the start of the abstract seems very abrupt. You need to add a topic sentence before the first sentence present in your abstract. Instead of starting with topic sentence, you have started your introduction with the evidence. 

From line 21 to 28, I suggest you just write name of study measures and remove what questionnaire and tools were used for assesing each study measure. I have rewritten as follow:

"In this survey,  data regarding COVID- 19 vaccination acceptance, fear, side effects, family members’ attitudes toward vaccination, as well as mental health status, neuropsychiatric symptoms, and cognitive impairments was collected among the elderly with MND."

Conclusion of the abstract need to be strengthen. Also add more study findings.

Introduction:

In the introduction, I will suggest you to write the rough estimate of death and COVID 19 cases. I am more interested in Case fatality rate. Write worldwide over X million of people have been affected by various strains of Corona virus, of which X% encounter deaths. Scientific writing should be meaningful to the reader.

Content of line 52-57 is not a good fit for defining the introduction. Remove it.

Remove this "30.26% of unvaccinated individuals tested positive for COVID-19, compared to 37.16% of vaccinated individuals"

The introduction is extremely lengthy. I suggest you to summarize as much as you can. Ideally, it should be of 500-800 words. In special cases, I am happy with an introduction of 1000 words. Summarize your introduction as much as you can. 

I have thoroughly read your introduction, and I found that most of the content discussed in your introduction should be part of your discussion, but it should be discussed with your study findings.

Methods:

Remove this: "The questionnaire-based study did not involve experiments on humans or human tissue samples."

Under the participants heading, I also want to know did you calculate the sample size. If you have not calculated the sample size, write in your study limitation. Also write sampling methods as study limitations.

In your study, most of the variables were continuous. Did you check the VIF and collinearity of each continuous variable? If not recheck it and remove those with high VIF value. Also, mentioned what cut-off value did you use for removing the collinearity? 

Results:

Mean should be written with plus and minus sign. use word Mean rather than average.

Remove this: "The participants’ mean score for concerns regarding the side effects of COVID-19 vaccines for their elderly families with MND was 1.4 (SD = 0.9). Furthermore, the mean level of perceived family support for vaccinating elderly families with MND was 3.0 (SD = 1.0), whereas 47.0% had received information about COVID-19 vaccines from social media. The participants’ mean scores for the four domains of the P-MoVac-COVID19S-12 ranged between 14.0 (SD = 3.8) and 17.0 (SD = 3.6). The mean score for caregiver intention to vaccinate their elderly families with MND against COVID-19 was 7.1 252 (SD = 3.1)." Rather than refer write name of all measure and refer to Table-1.

Split table 1 into 2: (1) Demographic profile of the participant, (2) Immunization behavior, attitude, and practices.

Content of line 262 to 274 should be summarized. I suggest you write all positively corelated behaviour and practices in one sentence, and in the next sentence write all the negatively corelated behavior and practices.

Remove this: "Caregivers’ perceived positive impacts of vaccination tended to positively correlate with caregiver intention (p = 0.054). Caregivers’ concern regarding side effects of COVID-19 vaccines and knowledge of vaccines were not significantly correlated with caregiver intention (p > 0.05)."

I am more interested to see Odds and Confidence interval, rather than p-value. I want to know the extent rather than Yes or no. 

Merge Table 2 and Table 3, and discuss only multivariate findings. No need to explain the bivariate findings.

Discussion:

Second paragraph of discussion is devoid of citation. I suggest you add some references and write discussion in light of other studies. 

Line 322 correct the spelling of present.

Spelling of present is not correct at other places.

Remove an extra p from word previous.

In line 330 to 337, you can add some content from your introduction section.

Last paragraph page 8 starting from line 338 need to be start from your study findings. Then discuss other studies findings.

Add a separate heading of study limitations: add more limitations such as generalisibility, sampling method, sample size, study design and study setting.

I am also interested in study strength. This is a good study, so you should have to write strengths too.

In short, I suggest you try to summarize the content esp of introduction, remove bivariate analysis (table 2) explanation, and intgrate discussion. Reduce the assumptions in discussion section.

Author Response

We appreciated your valuable comments. As discussed below, we have revised our manuscript with underlines based on your suggestions. Please let us know if we need to provide anything else regarding this revision.

Comment 1

The abstract is nicely written. I feel that the start of the abstract seems very abrupt. You need to add a topic sentence before the first sentence present in your abstract. Instead of starting with topic sentence, you have started your introduction with the evidence.

Response

Thank you for your comment. We added a topic sentence in the start of Abstract. Please refer to line 17-19.

Vaccination helps reduce the risk of coronavirus disease 2019 (COVID-19) infection in elderly individuals with major neurocognitive disorder (MND). However, some caregivers are hesitant to have their elderly family members with MND vaccinated against COVID-19.

Comment 2

From line 21 to 28, I suggest you just write name of study measures and remove what questionnaire and tools were used for assesing each study measure. I have rewritten as follow:

"In this survey, data regarding COVID- 19 vaccination acceptance, fear, side effects, family members’ attitudes toward vaccination, as well as mental health status, neuropsychiatric symptoms, and cognitive impairments was collected among the elderly with MND."

Response

We rewrote the sentence in Abstract accordingly. Please refer to line 21-24.

In this survey, data regarding COVID- 19 vaccination acceptance, fear, side effects, family members’ attitudes toward vaccination, as well as mental health status, neuropsychiatric symptoms, and cognitive impairments were collected among the elderly with MND.

Comment 3

Conclusion of the abstract need to be strengthen. Also add more study findings.

Response

We revised the conclusion in Abstract. Please refer to line 29-33.

This study demonstrated that caregiver factors (perceived family support, value of vaccination, and autonomy) and elderly families’ age were correlated with caregiver intention. These factors should be considered in developing interventions to enhance caregiver intention to vaccinate elderly families with MND against COVID-19.

Comment 4

Introduction:

In the introduction, I will suggest you to write the rough estimate of death and COVID 19 cases. I am more interested in Case fatality rate. Write worldwide over X million of people have been affected by various strains of Corona virus, of which X% encounter deaths. Scientific writing should be meaningful to the reader.

Response

Thank you for your suggestion. We totally agree that adding the percentage is needed. However, the WHO stopped calculating the percentage of mortality of COVID-19; thus, we did not have the accurate number. Instead, we added the numbers of excess mortality rates due to COVID-19 into Introduction. Please refer to line 42-44.

“Excess mortality rates exceeded 150 deaths per 100 000 population during at least one year of the COVID-19 pandemic in 80 countries and territories [2].”

Comment 5

Content of line 52-57 is not a good fit for defining the introduction. Remove it.

Response

We removed them. Please refer to line 48.

Comment 6

Remove this "30.26% of unvaccinated individuals tested positive for COVID-19, compared to 37.16% of vaccinated individuals"

Response

We removed them. Please refer to line 59.

Comment 7

The introduction is extremely lengthy. I suggest you to summarize as much as you can. Ideally, it should be of 500-800 words. In special cases, I am happy with an introduction of 1000 words. Summarize your introduction as much as you can. 

Response

Thank you for your suggestion. We summarized the Introduction into about 1000 words. Please refer to line 37-127.

Comment 8

I have thoroughly read your introduction, and I found that most of the content discussed in your introduction should be part of your discussion, but it should be discussed with your study findings.

Response

We added Discussion to containing the content mentioned in Introduction. We listed the changes in Responses to Comments 19, 21 and 22.

Comment 9

Methods:

Remove this: "The questionnaire-based study did not involve experiments on humans or human tissue samples."

Response

We removed them. Please refer to line 156.

Comment 10

Under the participants heading, I also want to know did you calculate the sample size. If you have not calculated the sample size, write in your study limitation. Also write sampling methods as study limitations.

Response

Thank you for your comment. We added the content introducing the calculation of the sample size into Methods section. Please refer to line 148-152.

According to Green [37], a total number of participants in the studies used a regression analysis model needs at least 50 + 8 * (the number of independent variables). There were 16 independent variables in this study; this study needed at least 178 participants. Therefore, 232 parents were enough for the regression analysis used in this study.

Comment 11

In your study, most of the variables were continuous. Did you check the VIF and collinearity of each continuous variable? If not recheck it and remove those with high VIF value. Also, mentioned what cut-off value did you use for removing the collinearity? 

Response

Yes, we checked the VIF and collinearity of each continuous variable. We listed the results below. The cut-off of 5/6 for the BSRS-5 is commonly used to indicate poor mental health status in Taiwan. Please refer to line 284-286.

The variance inflation factor of these seven variables ranged between 1.021 and 4.171, and the condition index was 29.623. The results indicated no collinearity, as determined on the basis of the suggestions of Senaviratna and Cooray [47].

Comment 12

Results:

Mean should be written with plus and minus sign. use word Mean rather than average.

Response

We revised them. Please refer to line 255-262.

The mean age of the participants was 58.1 ± 11.3 years. the mean age of their elderly families with MND was 81.5 ± 8.7 years…. Their mean severity of neuropsychiatric symptoms on the NPI-Q was 11.2 ± 7.6, and the mean severity of cognitive impairments on the CDR was 1.5 ± 0.5.

Comment 13

Remove this: "The participants’ mean score for concerns regarding the side effects of COVID-19 vaccines for their elderly families with MND was 1.4 (SD = 0.9). Furthermore, the mean level of perceived family support for vaccinating elderly families with MND was 3.0 (SD = 1.0), whereas 47.0% had received information about COVID-19 vaccines from social media. The participants’ mean scores for the four domains of the P-MoVac-COVID19S-12 ranged between 14.0 (SD = 3.8) and 17.0 (SD = 3.6). The mean score for caregiver intention to vaccinate their elderly families with MND against COVID-19 was 7.1 252 (SD = 3.1)." Rather than refer write name of all measure and refer to Table-1.

Response

We removed them and referred the results to Tables 1 and 2.

Table 1 presents the characteristics of the participants, and the characteristics and scores of CDR and NPI-Q of their elderly families with MND. Please refer to line 253-256.

“Table 2 presents the caregiver intention to vaccinate their elder families with MND, concerns regarding the side effects of COVID-19 vaccines, perceived family support for vaccinating elderly families with MND, received information about COVID-19 vaccines from social media, and the scores of the P-MoVac-COVID19S-12 and FCV-19S.” Please refer to line 262-266.

Comment 14

Split table 1 into 2: (1) Demographic profile of the participant, (2) Immunization behavior, attitude, and practices.

Response

Thank you for your suggestion. We split Table 1 into two tables (Tables 1 and 2). Please refer to line 267 and 270.

Comment 15

Content of line 262 to 274 should be summarized. I suggest you write all positively corelated behaviour and practices in one sentence, and in the next sentence write all the negatively corelated behavior and practices.

Response

We wrote them accordingly. Please refer to line 279-283.

The analysis revealed that caregiver perceived family support for vaccination (p < 0.001), perceived value of vaccination (p = 0.002), and autonomy to vaccinate elder families with MND (p = 0.001) were positively correlated with caregiver intention to vaccinate elderly families with MND, whereas elderly families’ age was negatively correlated with caregiver intention (p = 0.036).

Comment 16

Remove this: "Caregivers’ perceived positive impacts of vaccination tended to positively correlate with caregiver intention (p = 0.054). Caregivers’ concern regarding side effects of COVID-19 vaccines and knowledge of vaccines were not significantly correlated with caregiver intention (p > 0.05)."

Response

We removed them. Please refer to line 284.

Comment 17

I am more interested to see Odds and Confidence interval, rather than p-value. I want to know the extent rather than Yes or no. 

Response

Given that the associations of caregiver and elderly families with MND with caregiver intention were examined using bivariable and multivariable regression analysis, B, standard error and p value were shown in tables.

Comment 18

Merge Table 2 and Table 3, and discuss only multivariate findings. No need to explain the bivariate findings.

Response

Thank you for your suggestion. We merged Table 2 and Table 3 and discuss only multivariate findings. Please refer to line 272-279.

Table 3 presents the results of bivariable and multivariable linear regression analysis examining the factors related to caregiver intention to vaccinate their elderly families with MND against COVID-19. Factors that were significantly correlated with caregiver intention to vaccinate their elderly families with MND against COVID-19 in bivariable linear regression analysis models (caregiver perceived family support for vaccinating, four domains of motors to vaccinate elderly families with MND, concern regarding the side effects of COVID-19 vaccines, and elderly families’ age) were further entered into a multivariable linear regression analysis model (Table 3).

Comment 19

Discussion:

Second paragraph of discussion is devoid of citation. I suggest you add some references and write discussion in light of other studies. 

Response

We revised this paragraph by adding references and discussing in light of other studies. Please refer to line 306-317.

A previous study also found that perceived unfavorable family attitudes toward vaccinating their children was correlated with caregiver hesitancy to vaccinate their children [48]. According to the TPB [24,26], the favorable opinions of other family members can give caregivers a sense of the norm of the attitudes of those around them toward vaccination and reduce caregivers' hesitation to have elderly families with MND vaccinated against COVID-19. According to ecological systems theory [49], caring for elderly families with MND is an issue of interaction among caregivers, elderly families with MND, and other family members. Family members who support vaccination can also provide vaccination information to caregivers and get elderly families with MND vaccinated as soon as possible. Cues regarding vaccination from trusted people also affect caregiver COVID-19 vaccination acceptance [18].

Comment 20

Line 322 correct the spelling of present. Spelling of present is not correct at other places. Remove an extra p from word previous.

Response

Thank you for your reminding. We corrected the typos, and please refer to lines 344 and 346.

Comment 21

In line 330 to 337, you can add some content from your introduction section.

Response

We revised this paragraph by adding references and discussing in light of other studies. Please refer to line 331-341.

The present study also found that the positive impact motor of vaccination tended to be positively correlated with caregiver intention to vaccinate elderly families with MND. The items on the impact domain of the P-MoVac-COVID19S-12 asked caregivers their agreement with the effects of vaccination on protecting elderly families with MND against the COVID-19 and reducing their risk of catching COVID-19 [39]. According to the PMT [27–31], perceived efficacy of vaccination in alleviating the threat of COVID-19 influences individuals’ intention to receive vaccination against COVID-19. Caregiver who perceive greater impacts have a stronger belief in the differences made by vaccination uptake to prevent COVID-19 transmission; thus, their intention to vaccinate their elderly families with MND increases.

Comment 22

Last paragraph page 8 starting from line 338 need to be start from your study findings. Then discuss other studies findings.

Response

Thank you for your comment. We revised it accordingly. Please refer to line 352-361.

The present study did not find the significant associations of caregiver concerns regarding side effects of COVID-19 vaccines and fear of COVID-19, and MND individuals’ cognitive dysfunctions and neuropsychiatric symptoms with caregiver intention. The results of the present study were not congruent with those of the previous studies [18–20]. Studies have found that high caregiver concerns regarding side effects of COVID-19 vaccines and low caregiver fear of COVID-19 have been found to be correlated with low caregiver intention to vaccinate their elderly families with MND [18,19]; high cognitive and daily dysfunctions and low neuropsychiatric symptoms of agitation and aggression have been also found be correlated with vaccination against COVID-19 in elderly individuals with MND [20].

Comment 23

Add a separate heading of study limitations: add more limitations such as generalisibility, sampling method, sample size, study design and study setting.

Response

We added a heading and more limitations into the revised manuscript. Please refer to line 382 and 389-405.

“4.1. Strengths and Limitations

…This study has several limitations. First, the data collection relied solely on self-reports from caregivers of elderly individuals with MND, potentially introducing single informant bias and social desirability bias. Second, this study examined the cross-sectional associations between the factors and caregiver intention and thus did not establish causality. Third, whether this study’s results can be generalized to caregivers of elderly individuals with MND who have not sought medical assistance in outpatient settings remains uncertain. Further studies on caregivers of elderly individuals with MND in community are needed to replicate the results of this study. Fourth, caregivers of elderly individuals without MND were not recruited into this study for comparison. Further studies are warranted to examine whether the factors related to caregivers’ intentions identified in this study also relate to the intentions of caregivers of elderly individuals without MND. Fifth, the participants may also have provided socially desirable responses instead of choosing responses reflective of their true feelings. Potential information bias should be examined in further studies. Sixth, this study was conducted in 2024, well past the peak of COVID-19. However, as large-scale respiratory viral infectious disease outbreaks are still possible in the future, continued research is still necessary.

Comment 24

I am also interested in study strength. This is a good study, so you should have to write strengths too.

Response

Thank you for your positive comment. Please refer to line 382-388.

“4.1. Strengths and Limitations

The present study is the first one to comprehensively examine the roles of caregiver vaccination behaviors, attitudes, practices, sources of information and mental health status and elderly individuals’ neuropsychiatric symptoms and cognitive impairments on caregiver intention to vaccinate elderly families with MND against COVID-19. The results of this study provide evidence for developing intervention programs to enhance caregiver intention to vaccinate their elderly families with MND against COVID-19.

Comment 25

In short, I suggest you try to summarize the content esp of introduction, remove bivariate analysis (table 2) explanation, and integrate discussion. Reduce the assumptions in discussion section.

Response

We made revisions accordingly. We appreciate your valuable comments on our manuscript.

Reviewer 2 Report

Comments and Suggestions for Authors

This article is consider the main factors what motivate the caregivers  to vaccinate the elder families with major cognitive disorders. It is a well written article based on well validated researches.The associations between these variables and caregiver intention to vaccinate elderly families against COVID-19 were examined using a multivariable linear regression analysis model. It has been revealed  that caregiver first of all  perceived family support for vaccination as well  perceived value of vaccination, and autonomy to vaccinate elder families for their  positive solution.  In overall it is very actual finding and might be accept to publication in Vaccines.

Author Response

Comment

This article is consider the main factors what motivate the caregivers to vaccinate the elder families with major cognitive disorders. It is a well written article based on well validated researches. The associations between these variables and caregiver intention to vaccinate elderly families against COVID-19 were examined using a multivariable linear regression analysis model. It has been revealed that caregiver first of all perceived family support for vaccination as well perceived value of vaccination, and autonomy to vaccinate elder families for their positive solution. In overall it is very actual finding and might be accept to publication in Vaccines.

Response

Thank you for your positive comment.

Reviewer 3 Report

Comments and Suggestions for Authors

§  Writing COVID-19 would be better instead of “Corona- 3 virus Disease 2019” in the title.

§  Moreover, the title may be revised to look better in terms of conciseness and grammar.

§  The major and minor side effects of the vaccines can briefly be mentioned. What about the recently acknowledged platelet-related side effects of Covishield?

§  Why were specifically elderly populations chosen?

§  If the vaccines are rarely used anymore, then what is the significance of the content?

§  Why the group aged 20 years or older was included if it is done for the elderly population?

Comments on the Quality of English Language

Moderate improvements is required.

Author Response

Comment 1

  • Writing COVID-19 would be better instead of “Corona- 3 virus Disease 2019” in the title.

Response

Thank you for your suggestion. We revised it accordingly. Please refer to line 4.

Comment 2

  • Moreover, the title may be revised to look better in terms of conciseness and grammar.

Response

We revised the title into “Factors Related to Caregivers’ Intention to Vaccinate Elderly Their Families with Major Neurocognitive Disorder Against COVID-19.” Please refer to line 2-4.

Comment 3

  • The major and minor side effects of the vaccines can briefly be mentioned. What about the recently acknowledged platelet-related side effects of Covishield?

Response

We added the introduction for the major and minor side effects of COVID-19 vaccines in Introduction section. Please refer to line 85-90.

Pain at the injection site, fever, and fatigue were the most common side effect in people receiving COVID-19 vaccines [21]. Major adverse effects of COVID-19 vaccines such as cardiovascular, neurologic, hematologic, and immune-allergic side effects have been also reported [22]. For example, several vaccine-induced thrombotic thrombocytopenia syndrome cases have been reported after the ChAdOx1 nCov-19 vaccination [23].

Comment 4

  • Why were specifically elderly populations chosen?

Response

The present study focused on elderly population with major neurocognitive disorder for two reasons. First, elderly individuals have been identified as the population with a high mortality rate of COVID-19, severe COVID-19 symptoms and complications, and the need of invasive mechanical ventilation. Second, elderly individuals with the diagnosis of major neurocognitive disorder have increased COVID-19 severity and mortality. We introduced the reason in Introduction section. Please refer to line 44-50.

Comment 5

  • If the vaccines are rarely used anymore, then what is the significance of the content?

Response

We added the importance of this study into Introduction section. Please refer to line 71-76.

Although the World Health Organization declared on 3 May 2023 that COVID-19 was no longer a public health emergency of international concern, the numbers of confirmed COVID-19 cases and COVID-19-associated deaths worldwide have continued to increase. Therefore, examining the factors related to caregiver intention to vaccinate their elderly families with MND is essential to the development of intervention programs for enhancing the rate of vaccination in elderly individuals with MND.

Comment 6

  • Why the group aged 20 years or older was included if it is done for the elderly population?

Response

This study examined caregivers’ intention to vaccinate their elderly families with major neurocognitive disorder. Therefore, caregivers who aged 20 years or older were invited into this study.

Reviewer 4 Report

Comments and Suggestions for Authors

Dear authors, thank you for allowing me to review this interesting manuscript, which assesses the factors associated with caregiver intention to vaccinate elderly families with MND.

Although I found the manuscript well-written and methodologically sound, I have identified some points of concern that should be addressed by the authors to improve the overall quality of this interesting manuscript.

INTRODUCTION

This clear section drives the river to the main points (research problem, state of the art, rationale, and hypotheses).

METHODS

Are clearly described but could benefit from a description of the context in which the caregivers were included.

RESULTS

Are clear and well-described.

DISCUSSION

This is the section that raised more concerns, as for the first part discussing the perceived family support totally lacks a citation. Moreover, large attention has been driven to the prioritization of frail people in the vaccination campaign that is already a solution. Lastly, in the limitations section there is not mention that the study has been conducted in 2024, which is a little bit out of time for the COVID-19 vaccination campaign. This should be disclosed and discussed.

Please, check for typos in this section, as there are some words that are misspelt. (e.g., fmailies, uptak, presetn, ...)

CONCLUSION

This section totally lacks of a part in which interventions to be introduced are clearly described; moreover, some research perspectives are needed.

I hope these suggestions will help to improve the overall quality of this manuscript. 

Author Response

We appreciated your valuable comments. As discussed below, we have revised our manuscript with underlines based on your suggestions. Please let us know if we need to provide anything else regarding this revision.

Comment 1

INTRODUCTION

This clear section drives the river to the main points (research problem, state of the art, rationale, and hypotheses).

Response

Thank you for your positive comment.

Comment 2

METHODS

Are clearly described but could benefit from a description of the context in which the caregivers were included.

Response

Thank you for your suggestion. We added a description of the context in which the caregivers were included. Please refer to line 138-144.

Psychiatrists told caregivers, “This is a questionnaire study to understand caregivers’ intention and correlates to have their elderly family members with MND vaccinated against COVID-19. The results of the study will serve as a reference for clinicians to develop strategies to enhance caregivers’ efforts to have their elderly family members with MND vaccinated. The results of the questionnaire are confidential. You can be assured that if you do not want to participate in the study, it will not affect the medical rights of your family members.

Comment 3

DISCUSSION

This is the section that raised more concerns, as for the first part discussing the perceived family support totally lacks a citation. Moreover, large attention has been driven to the prioritization of frail people in the vaccination campaign that is already a solution. Lastly, in the limitations section there is not mention that the study has been conducted in 2024, which is a little bit out of time for the COVID-19 vaccination campaign. This should be disclosed and discussed.

Response

We revised the manuscript based on your comments.

  1. We revised the first paragraph of Discussion section and added new references for it. Please refer to line 306-317.

“…A previous study also found that perceived unfavorable family attitudes toward vaccinating their children was correlated with caregiver hesitancy to vaccinate their children [48]. According to the TPB [24,26], the favorable opinions of other family members can give caregivers a sense of the norm of the attitudes of those around them toward vaccination and reduce caregivers' hesitation to have elderly families with MND vaccinated against COVID-19. According to ecological systems theory [49], caring for elderly families with MND is an issue of interaction among caregivers, elderly families with MND, and other family members. Family members who support vaccination can also provide vaccination information to caregivers and get elderly families with MND vaccinated as soon as possible. Cues regarding vaccination from trusted people also affect caregiver COVID-19 vaccination acceptance [18].…”

  1. We added a new paragraph in Discussion section to discuss the level of caregiver intention to vaccinate their elderly families with MND against COVID-19 because about 10% of caregivers rated their intention as very low. Please refer to line 296-303.

“In this study, caregivers expressed a moderate to high level of intention to vaccinate their elderly families with MND against COVID-19, with a mean score of 7.1 on a 10-point Likert scale. However, some caregivers remained hesitant to have their elderly families with MND vaccinated against COVID-19, as evidenced by 24 (10.3%) participants who rated their intention to vaccinate as very low (score of 1). Considering the benefits of COVID-19 vaccination, educating caregivers about its importance and strengthening their intention to have their elderly families with MND vaccinated is essential.

  1. We listed it as one of the limitations of this study. Please refer to line 403-405.

Sixth, this study was conducted in 2024, well past the peak of COVID-19. However, as large-scale respiratory viral infectious disease outbreaks are still possible in the future, continued research is still necessary.

Comment 4

Please, check for typos in this section, as there are some words that are misspelt. (e.g., fmailies, uptak, presetn, ...)

Response

Thank you for your reminding. We corrected the typos, and please refer to lines 323, 327, 331 and 344.

Comment 5

CONCLUSION

This section totally lacks of a part in which interventions to be introduced are clearly described; moreover, some research perspectives are needed.

Response

Thank you for your comment. We revised the Conclusion section and added introduction for interventions and further issues warranted study. Please refer to line 410-422.

Based on the results of this study, healthcare professionals must formulate strategies to enhance intention among caregivers to vaccinate their elderly families with MND against COVID-19. Priority should be placed on intervention programs targeting caregivers who have older families with MND. Healthcare professionals must understand what attitudes of other family members toward vaccination caregivers perceive and assist them in shaping the ability to communicate the necessity of vaccination with other family members. Moreover, these findings underscore the requirement for programs designed to enhance caregivers’ motors of vaccination such as value, positive impacts and autonomy to increase their intention to vaccinate their elderly families with MND against COVID-19. Further studies are needed to examine caregiver intention and related factors to vaccinate elderly families with MND in the community population. The sources of information regarding COVID-19 and vaccination also warrant examination.

Round 2

Reviewer 4 Report

Comments and Suggestions for Authors

Dear Authors,

thank you for allowing me to review this second version of your interesting manuscript.

I would like to congratulate you on the improvements you made. You have carefully considered all my suggestions, and I am satisfied with the amendments provided. If I could suggest a final refinement in the editorial process, during the correction of the article draft, I would correct the verbs you used in the conclusion (must) with "should".